Novel transfer learning approach for hand drawn mathematical geometric shapes classification

Alam Aneeza 1
http://orcid.org/0000-0001-5429-9835 Raza Ali 2 ali.raza.scholarly@gmail.com
Thalji Nisrean 3
Abualigah Laith 4 5 6 7
http://orcid.org/0000-0003-0101-4781 Garay Helena 8 9 10
Alemany-Iturriaga Josep 10 11 12
http://orcid.org/0000-0002-8271-6496 Ashraf Imran 13 imranashraf@ynu.ac.kr
1 Institute of Computer Science, Khwaja Fareed University of Engineering and Information Technology , Rahim Yar Khan , Pakistan
2 Department of Software Engineering, University of Lahore , Lahore , Pakistan
3 Faculty of Computer Studies, Arab Open University , Amman , Jordan
4 Computer Science Department, Al al-Bayt University , Mafraq , Jordan
5 Chitkara University Institute of Engineering and Technology, Centre for Research Impact & Outcome , Rajpura, Punjab , India
6 Applied Science Private University, Applied Science Research Center , Amman , Jordan
7 University of Tabuk, Artificial Intelligence and Sensing Technologies (AIST) Research Center , Tabuk , Saudi Arabia
8 Isabel Torres, Universidad Europea del Atlantico , Santander , Spain
9 Cuito, Universidade Internacional do Cuanza , Bie , Angola
10 Isabel Torres, Universidad de La Romana , La Romana , Dominican Republic
11 Universidad Internacional Iberoamericana, Universidad Internacional Iberoamericana , Campeche , Mexico
12 Universidad Internacional Iberoamericana Arecibo, Universidad Internacional Iberoamericana Arecibo , Puerto Rico , United States
13 Department of Information and Communication Engineering, Yeungnam University , Gyeongsan , Republic of South Korea
Alatas Bilal
Electronic publication date: 2025 Jan 31
Publication date: 2025
Volume: 11
Electronic Location ID: e2652
Received 2024 Jul 31; Accepted 2024 Dec 20
Copyright: © 2025 Alam et al.
Copyright year: 2025
Copyright holder: Alam et al.
License: This is an open access article distributed under the terms of the Creative Commons Attribution License, which permits unrestricted use, distribution, reproduction and adaptation in any medium and for any purpose provided that it is properly attributed. For attribution, the original author(s), title, publication source (PeerJ Computer Science) and either DOI or URL of the article must be cited.
License URL: https://creativecommons.org/licenses/by/4.0/

Keywords: Mathematical shapes, Transfer learning, Deep learning, Computer vision

Funding: European University of Atlantic This research is supported by the European University of Atlantic. The funders had no role in study design, data collection and analysis, decision to publish, or preparation of the manuscript.

==============================
Hand-drawn mathematical geometric shapes are geometric figures, such as circles, triangles, squares, and polygons, sketched manually using pen and paper or digital tools. These shapes are fundamental in mathematics education and geometric problem-solving, serving as intuitive visual aids for understanding complex concepts and theories. Recognizing hand-drawn shapes accurately enables more efficient digitization of handwritten notes, enhances educational tools, and improves user interaction with mathematical software. This research proposes an innovative machine learning algorithm for the automatic classification of mathematical geometric shapes to identify and interpret these shapes from handwritten input, facilitating seamless integration with digital systems. We utilized a benchmark dataset of mathematical shapes based on a total of 20,000 images with eight classes circle, kite, parallelogram, square, rectangle, rhombus, trapezoid, and triangle. We introduced a novel machine-learning algorithm CnN-RFc that uses convolution neural networks (CNN) for spatial feature extraction and the random forest classifier for probabilistic feature extraction from image data. Experimental results illustrate that using the CnN-RFc method, the Light Gradient Boosting Machine (LGBM) algorithm surpasses state-of-the-art approaches with high accuracy scores of 98% for hand-drawn shape classification. Applications of the proposed mathematical geometric shape classification algorithm span various domains, including education, where it enhances interactive learning platforms and provides instant feedback to students.

Introduction

Hand-written data is considered the most natural way of recording information (Jabde et al., 2024). Hand-drawn mathematical images refer to the sketches of geometric figures such as circles, squares, etc, created manually on a article or digital interface by hand (Duvernoy, 2024). These images or sketches are often rough and imperfect but are a quick way to convey mathematical concepts.

Image recognition presents a significant challenge in the field of computer vision (Wang et al., 2024). This area of research is important in educational technologies, mathematical tools, and digital notetaking applications, as the correct interpretation of user input can enhance the functionality and user experience. Traditional approaches to shape recognition rely on pattern recognition which can generate ambiguous results (Belongie, Malik & Puzicha, 2002). So, there is a need for more robust and adaptable methods that accurately anticipate these shapes regardless of drawing style, orientation, and scale. To bridge the gap between human input and digital processing, hand-drawn geometric shape recognition is necessary. Hand-drawn images are diverse and usually have uneven strokes, varying angles, and inconsistent degrees of closure. In this context, machine learning algorithms offer great preciseness.

Applications of hand-drawn mathematical geometric shape recognition spread across various fields (Alwaely & Abhayaratne, 2019). In education, it increases interactive learning platforms, allowing real-time feedback and assessment. In design and engineering, it helps in converting initial sketches into precise digital models, streamlining the design process (Fang, Feng & Cai, 2022). Moreover, this technology can be integrated into mobile applications and software tools that require user-friendly interfaces for geometric input, making mathematical problem-solving more accessible and efficient.

This research adds to the literature by proposing a novel machine-learning algorithm that gives the most optimized results during evaluation. We also checked how different machine-learning algorithms perform on this publicly available dataset. These contributions would advance the field by improving recognition capabilities and expanding the understanding of how different algorithms perform on hand-drawn shapes.

The following are the primary research contributions of the current research: This study introduced a novel approach called CnN-RFc that uses the convolutional neural network (CNN) for spatial feature extraction. The extracted features are later used with the random forest classifier (RFC) to obtain probabilistic features, which are used to train machine learning models.

Five advanced machine learning algorithms in comparison are used for performance comparison. Hyperparameter optimization and cross-variation approaches are applied to enhance models’ performance and generalization.

Several state-of-the-art approaches from existing literature are also considered for performance comparison. In addition, an ablation study is also carried out for further validation.

This research article is divided into seven sections. The “Introduction” gives a basic introduction to the research, which covers answers to questions like why recognition of mathematical shapes is important and how it is helpful. “Literature Review” contains a review of previous studies conducted on this topic or similar research. “Proposed Methodology” discusses the methodology proposed to train machine-learning algorithms. It includes the dataset used, pre-processing techniques, feature extraction, applied ML algorithms, parameter tuning, and results optimization steps. In “Experiments and Discussions”, the results analysis and discussion portion are included. The conclusions and future work are in “Conclusion and Future Directions”, which gives an overview of the future prospects of the study.

Literature review

It is evident from past studies that hand-written data is more logical and easy to learn (Alonso, 2015). Handwritten data increases the ability to focus and improves information organization and prioritizing skills. It is proved that the solution to a conceptual problem is easy to understand when practiced by hand (Mueller & Oppenheimer, 2014). It shows how important it is to take handwritten notes or lectures. It is hard to handle these physical data records. Moreover, it takes more space and cost and is also unchangeable. Compared to these handwritten documents, digital records are easy to manage and manipulate when needed. That is why handwritten document recognition techniques have been significant for scholars.

Various writers studied shape recognition and prediction. The study by Hse & Newton (2004) presented an online system for hand-sketched symbols. They used support vector machine (SVM), nearest neighbor (NN), and minimum mean distance (MMD) for the classification of image type. A dataset of 7,410 hand-sketched images is used and predicted with 97% accuracy using Zernick moments features. The HHReco dataset containing 7,791 samples of PowerPoint symbols used by the researcher achieved 98.2%, 96.7%, and 95.1% accuracy using image deformation model (IDM), Zernike Moments, and Feature neural networks, respectively (Ouyang & Davis, 2009).

A study by Fu & Kara (2011) utilized CNN as a recognizer along with two localization methods to recognize hand-drawn engineering diagrams and achieved an accuracy of 97.7%. They used a dataset containing 328 images in 16 categories and then enhanced it to 400 images per category using random local distortions. Another study by Eicholtz & Kara (2015) used SVM and the multiobject evolutionary model to recognize mechanical linkages in textbook images and sketches. The study achieved an accuracy of 73% for recognizing mechanical joints in human-drawn sketches. Similarly, CNN trained on offline features of 101 classes of hand-drawn mathematical shapes achieved an accuracy of 91.88% (Ayeb, Meguebli & Echi, 2021).

In the work by Wang et al. (2016), the recognition of hand-drawn flowcharts was studied. The flowcharts were considered as a two dimensional language with two elements i.e., symbols and arrows. The data process was divided into three steps. First, symbol hypothesis generation was performed. Second, the symbols were recognized. Third, layout interpretation uses grammar details to structure the information at the text level.

The study Roy et al. (2020) was conducted to identify the components in an electrical circuit diagram. The dataset consisted of twenty circuit components with each class having 150 images making a total of 3,000 images of hand-drawn circuit portions. The components were classified using a minimal sequential optimization classifier. The experimental outcome concluded that the proposed approach provided an average of 93.83% accuracy on the dataset.

In 2021, the study by Dey et al. (2021), Fang, Feng & Cai (2022) used 150 hand-drawn circuit images of 64 × 64 pixels. They proposed a customized convolution neural network to predict electric circuit components. They achieved an accuracy of 97.3%. The researchers proposed building a high-order Markov random field on the stroke level to handle segmentation and recognition simultaneously. The training dataset comprised 5,540 symbols for flowchart (FC) sketches and 3,631 for finite automata (FA) diagrams. The accuracy achieved by the study was 84.4% for flowcharts and 95.8% for finite automata diagrams. A recent study conducted by Fang, Feng & Cai (2022) improved the performance by 2.5% and 1.7% for FC and FA, respectively, compared to earlier state-of-the-art studies.

In Alfano et al. (2024), a transfer learning-based method was proposed for image classification with fast kernel methods to fine-tune. They used data from 3,500 training samples for the process. The study results were fine. Similarly, in Öztürk-Birim & Gündüz-Cüre (2024), a transfer learning approach was presented for the classification of electronic waste images. They have used a combination of CNN and ImageNet as transfer which have shown promising results. In addition, in another research (Jeong & Moon, 2024), an adaptive transfer learning method was introduced for double random phase encoding using complex datasets based on road surveillance camera images.

Hand-drawn sketches are handy for collaborative communication and real-time but still, there is sometimes a need to transform them into digital form for further processing. Many advanced techniques are used to address this need, but most are field-specific, limiting the user from other backgrounds to utilize them. Then the Flowmind2digital method was proposed, which used neural network architecture design and keypoint detection approach to improve the overall accuracy of recognition. The dataset used consisted of 1,776 hand-drawn flowcharts annotating 22 scenarios. They achieved an accuracy of 87.3% (Liu et al., 2023). Digital conversion of hand-drawn images or circuits is not a new concept. In 2022, a study conducted by Rachala & Panicker (2022) utilized the YOLOv5 method to detect the components of an electronic circuit. They succeeded by achieving an accuracy of 98.2% (Rachala & Panicker, 2022).

Research gap

Current algorithms struggle to effectively generalize different drawing styles, orientations, and variations in sizes (El-Kenawy et al., 2024; Abdollahzadeh et al., 2024). Novel machine-learning approaches, explainable models, and optimization for low-power devices are needed. There is also a lack of research considering hand-drawn shapes, as well as adaptive learning systems that personalize recognition over time. Little literature is found on hand-drawn flowcharts and finite automated images, which pave the way for more studies on these topics. The summary of the analyzed literature is provided in Table 1.

Table 1 Literature review.

Ref.	Year	Dataset size	Method used	Accuracy	
Hse & Newton (2004)	2004	7,410 hand-sketched images	SVM, MMD, NN	97%	
Ouyang & Davis (2009)	2009	7,791 Powerpoint symbols	IDM, Zernick moments, FNN	98%	
Wang et al. (2016)	2016	9,171 diagrams	High order Markov model field	95%	
Roy et al. (2020)	2020	3,000 circuit components	SMO classifier	93%	
Dey et al. (2021)	2021	150 circuit images	CNN	97%	
Fang, Feng & Cai (2022)	2022	5,540 flowchart symbols 3,631 FA diagrams	CNN	95%	
Rachala & Panicker (2022)	2022	154 circuit images	YOLOv3, YOLOv5, SSD300	98%	
Liu et al. (2023)	2023	1,776 flowchart diagrams	NN, Key point detection	87%	

Proposed methodology

This section gives details of proposed methods availed for the recognition of mathematical shapes. We have clearly outlined the image dataset used to develop classification algorithms. Figure 1, gives the step-wise working flow of the proposed methodology. First, the dataset is obtained from a publicly available data repository comprising images of hand-drawn mathematical shapes. The dataset is then checked to see if it is properly formatted for further analysis. Then feature extraction is performed using Conv2D and MaxPooling operations over 256 × 256 × 3 size images. The extracted dataset is then divided into training and test datasets with an 80:20 proportion, respectively. After the train-test split, classical and proposed models are trained on train data their performance is evaluated against the test dataset. Hyperparameter tuning further refined the models to improve their efficiency and accuracy. The model demonstrating superior performance through the validation process was then employed for mathematical shape recognition. This extensive methodology aims to use the capabilities of machine learning algorithms to detect mathematical hand-drawn shapes.

Figure 1 The workflow of the study to recognize hand-drawn mathematical images using the proposed technique.

Mathematical geometric shapes data

The shapes image dataset used in this study is taken from an open-access data repository i.e., Kaggle (Rivaldo, 2024). It is constructed with eight classes: “Circle”, “Kite”, “Parallelogram”, “Square”, “Rectangle”, “Rhombus”, “Trapezoid”, and “Triangle”. Each class has 1,500 training samples making up a total of 20,000 image samples of mathematical shapes. Each sample is an image measuring (256 × 256 × 3) (RGB). The data set is cleaned and arranged in separate training, validation, and testing folders. A sample of dataset records is illustrated in Fig. 2.

Figure 2 An in-depth analysis of a sample dataset of hand-drawn mathematical shapes.

Ethical implications

In considering the ethical implications of this research, it is important to address the use of hand-drawn data in the context of data privacy. The dataset consists solely of drawings of shapes, which inherently do not contain any personal or sensitive information. This characteristic significantly mitigates privacy concerns, as the absence of identifiable data means that individuals cannot be traced or recognized through the dataset.

Image pre-processing

To ensure that the dataset is in standard form and properly formatted, we ran some initial analysis. First is ensured that each image has the same dimensions such as 256 × 256 × 3. Then we labeled encode the data for training, giving each class a number as ‘circle’: 0, ‘kite’: 1, ‘parallelogram’: 2, ‘rectangle’: 3, ‘rhombus’: 4, ‘square’: 5, ‘trapezoid’: 6, ‘triangle’: 7. Then to check the distribution of each class, statistical analysis is performed which proved that the dataset is perfectly balanced. Figure 3 shows the count of each class in the training dataset.

Figure 3 The number of images per class in hand-drawn images dataset.

No data augmentation technique is used for this study. The dataset already contains a total of 20,000 image samples of various mathematical shapes, which are enough to train machine learning models. In addition, class distribution is balanced so there is no need for data augmentation.

Novel transfer feature engineering algorithm: CnN-RFc

In this study of mathematical hand-drawn shape classification, we introduced a novel machine-learning method CnN-RFc that uses CNN for spatial feature extraction and the CnN-RFc for probabilistic feature extraction from image data. The architecture of the RFC classifier is illustrated in Fig. 4. This method uses transfer learning to improve feature generation, making it better than traditional techniques. First, the CnN-RFc method uses the CNN architecture to process an image dataset. CNNs are great at extracting spatial features from images, which helps identify complex patterns and details in hand-drawn image data. Then, these spatial features are fed into the RFC, which is excellent at generating class probabilistic features from data. By using RFC to refine these spatial features, RFC creates highly effective features for classifying mathematical shapes with remarkable accuracy. This research shows that the CnN-RFc method significantly outperforms the classical methods in image classification and recognition. In addition, during the features extraction of the approach, the layer-wise feature generations are illustrated in Fig. 5.

Figure 4 The step-wise mechanism of transfer features generations from hand-drawn mathematical shapes using the novel proposed CnN-RFc method.

Figure 5 The process of proposed transfer feature extraction of thermograms layer using a one-sample image from the dataset.

Applied artificial intelligence algorithms

Artificial intelligence (AI) methods for hand-drawn mathematical geometric shape recognition using image data have shown significant advancements, leveraging a combination of computer vision and machine learning approaches. CNNs are particularly effective for this task due to their capacity to learn spatial hierarchies of features from input images automatically. Evaluations of these AI models typically involve various metrics demonstrating their effectiveness in accurately identifying and classifying various geometric shapes.

2D-convolutional neural network

A 2D-CNN is a powerful model for recognizing hand-drawn mathematical geometric shapes from image data (Ahlawat et al., 2020). The network operates by applying a series of convolutional filters to the input images, which allows it to capture and learn the spatial hierarchies and features of the shapes. In the initial layers, the filters detect basic edges and textures, while deeper layers identify more complex structures and patterns characteristic of specific geometric shapes. The CNN can generalize well to new, unseen hand-drawn shapes, accurately classifying them into predefined categories such as circles, triangles, squares, and other geometric figures. The ability of CNNs to automatically learn and extract relevant features makes them highly effective for this task, outperforming traditional image processing techniques that rely on manually crafted features.

Random forest

A random forest classifier (RFC) (Thenuwara & Nagahamulla, 2017) is a powerful ensemble learning method, particularly effective for the task of recognizing hand-drawn mathematical geometric shapes using spatial features of image data. This technique builds many decision tree data during the training process and merges their prediction outputs to improve classification accuracy and control overfitting. For recognizing geometric shapes, the classifier utilizes spatial features extracted from image data, such as edges, contours, and moments, which describe the geometry and distribution of pixel intensities. These features are fed into the RFC, where each decision tree independently learns decision rules to distinguish between different shapes like circles, triangles, and squares. For a given dataset x, the classification output y^ can be expressed as:

(1) y^=majority_vote (T1(x),T2(x),…,Tn(x))

Here the Ti(x) represents the prediction of the ith decision tree in the forest. n is the total number of decision trees. majority_vote denotes the majority voting function. The final classification decision y^ is the class label.

Light gradient boosting machine

The Light Gradient Boosting Machine (LGBM) has proven to be an effective model for recognizing hand-drawn mathematical geometric shapes by leveraging spatial features of image data. The training begins with extracting spatial features from the image data and input to LGBM. The LGBM classifier then utilizes a gradient-boosting framework to improve its predictions iteratively. It constructs an ensemble of decision trees, where each tree focuses on correcting the errors of its predecessor. This method enhances the model’s ability to distinguish between different geometric shapes by learning intricate patterns and relationships within the spatial features. In the LGBM model, the prediction for a classification task can be represented as follows:

(2) y^=σ(∑m=1Mfm(x)).

Here y^ is the predicted probability of the positive class. σ(⋅) is the sigmoid function. M is the total number of trees. fm(x) represents the output of the m-th tree given input features x.

K-neighbors classifier

The K-neighbors classifier (KNC) is a non-parametric, instance-based learning algorithm that can be effectively utilized for recognizing hand-drawn mathematical geometric shapes through spatial features of image data. During the training phase, the KNC algorithm stores all available labeled examples, mapping each feature vector to its corresponding shape label. When a new, unlabeled image is encountered, the algorithm calculates the Euclidean distance between the feature vector of this new image and those of the stored examples. It then identifies the ‘k’ nearest neighbors, which are the training examples with the smallest distances to the new image. The class target label that appears most commonly among these neighbors is assigned to the new image. The classification function for the KNC can be written as:

(3) f(x)=arg⁡maxc∈C∑i=1kδ(yi,c).

Here x is the new data points. yi is the label of the i-th nearest neighbor. C is the set of all possible classes. δ(yi,c) is an indicator function. f(x) is the predicted class for x, based on the majority class of its k nearest neighbors.

Logistic regression

Logistic regression (LR) (Rahmatinejad et al., 2024) is a powerful statistical model often employed for classification tasks, including hand-drawn mathematical geometric shape recognition. The working of LR in this context involves using spatial features extracted from image data. Initially, hand-drawn shapes such as circles, squares, and triangles are digitized and preprocessed to enhance the quality of the input images. Key spatial features, such as edges, corners, and pixel intensities, are then extracted using edge detection and contour analysis techniques. These features serve as inputs for the LR model, which aims to distinguish between different geometric shapes. LR classification equation:

(4) P(y=1|x)=11+e−(β0+β1x1+β2x2+…+βnxn).

Here P(y=1|x) is the probability of the binary outcome y being 1 given the features x=(x1,x2,…,xn). β0 is the intercept term. β1,β2,…,βn are the coefficients for each feature x1,x2,…,xn.

Hyperparameter tuning

Hyperparameter tuning is a significant step in optimizing machine learning algorithms, enhancing their performance and predictive power for mathematical shape recognition. Fine-tuning these hyperparameters allows machine learning models to achieve generalization capabilities, essential for robust and reliable predictions in diverse applications. The best-fit hyperparameters are shown in Table 2.

Table 2 Hyperparameter tuning of machine learning algorithms.

Technique	Hyperparameters description	
RFC	n_estimators = 5, random state = 0, max_depth = 6, criterion = ‘entropy’	
LGBM	n_estimators = 40, num_leaves = 31, importance_type = ‘split’	
LR	max_iter = 10, multi_class = ‘deprecated’, C = 1.0	
KNC	n_neighbors = 9	
CNN	activation = ‘softmax’, loss = ‘categorical_crossentropy’, optimizer = adam, epochs = 10, Trainable params: 4,065,160 (15.51 MB)	

The RFC and CNN models underwent a systematic grid search and manual fine-tuning. For the RFC, the n_estimators = 5 parameter, we conducted a grid search over a range of values (e.g., 5, 10, 50, 100). We observed that n_estimators = 5 provided a balance between computational efficiency and classification accuracy on our dataset. We tested both ‘gini’ and ‘entropy’ as splitting criteria. The ‘entropy’ criterion, which considers information gain, showed slightly better performance on our dataset by improving class separability. For the CNN model, the softmax activation function is used in the output layer to handle the multi-class classification problem. The ‘categorical_crossentropy’ loss function is selected as it is well-suited for multi-class classification problems and works seamlessly with softmax activation. The Adam optimizer was chosen for its adaptive learning rate mechanism, which helps converge faster.

Experiments and discussions

The section presents the results analysis and discussions on a novel invented approach that utilizes transfer learning-based feature extraction in hand-drawn mathematical shape recognition. This section critically analyzes the performance measurement scores to evaluate the results and effectiveness of the introduced method compared to existing approaches.

Experimental setup

The experimental setup for this research is designed using a cloud-based Python IDE i.e., Jupyter Notebook, specifically Google Colab, with 90 GB of disk space and RAM of 13 GB. The dataset is divided into training and test datasets with an 80:20 proportion, respectively. After the train-test split, models are trained on training data, and their performance is evaluated against the test dataset. We evaluate the performance of the used machine learning technologies using performance metrics like accuracy (A), precision (P), recall (R), and F1 score.

Results with CNN

The outcomes of the CNN training on the shapes dataset are displayed in Fig. 6. The CNN model underwent training on image data over the course of 10 epochs. Initially, the training loss was substantial during the first two epochs, but it diminished to zero after that. Conversely, the training accuracy showed a steady increase. In the beginning, the accuracy scores for both training sets and validation sets were low, but they improved as the number of epochs increased. The findings reveal that CNN’s training accuracy consistently ranged between 85% and 95% throughout the training process.

Figure 6 The time series graph-based results analysis of applied CNN model.

Performance analysis with proposed features

Table 3 describes the describes the performance measurement of utilized machine learning algorithms. The proposed innovative methodology improved the results of the applied machine learning approaches. The majority of the techniques applied achieved accuracy scores exceeding 97%. Notably, the KNN and LGBM methods outperformed others, attaining the highest accuracy scores of 98%. This indicates that the proposed approach led to high-performance results in recognizing hand-drawn mathematical shapes.

Table 3 The comparison analysis of applied approaches.

Method	A	P	R	F1	
LGBM	0.98	0.97	0.981	0.98	
KNC	0.98	0.98	0.982	0.98	
LR	0.97	0.97	0.977	0.97	
RF	0.97	0.97	0.963	0.97	

The results we have obtained from the proposed novel LGBM algorithm are given in Table 4. It is seen that the proposed approach has successfully recognized the eight different classes of mathematical shapes. The precision and recall for each class is greater than 0.97, which means the model has generalized well over the mathematical shapes dataset. The proposed LGBM approach is well suited for the recognition of mathematical hand-drawn shapes.

Table 4 Classification by proposed LGBM algorithm.

Shape class	P	R	F1	
Circle	0.97	0.98	0.97	
Kite	0.97	0.98	0.98	
Parallelogram	0.98	0.98	0.98	
Rectangle	1.00	1.00	1.00	
Rhombus	0.97	0.98	0.98	
Square	0.99	0.98	0.98	
Trapezoid	0.99	0.98	0.98	
Triangle	0.98	0.97	0.97	

The outcomes of the performance assessment, utilizing the confusion matrix for the applied methods with the suggested features, are depicted in Fig. 7. The research indicates that these techniques attained lower error rates, thereby decreasing the misclassification rate during the unseen testing part. The findings demonstrate the enhanced performance of the introduced KNC approach, which exhibited a higher correct classification rate and a lower error rate in comparison with other methods.

Figure 7 Applied methods confusion matrix testing results analysis with the proposed approach.

K-fold validations analysis

The performance of the applied machine learning methods is evaluated using K-fold cross-validation, with the outcomes presented in Table 5. For validation, 10 data folds are employed. The analysis reveals that all methods achieved over 97% accuracy with high K-fold, accompanied by minimal standard deviation (SD) scores. This demonstrates the proposed KNC model’s effectiveness in generalizing for the recognition of hand-drawn mathematical shapes.

Table 5 The performance valuations based on k-fold cross-validation.

Method	Kfold	A	SD (+/−)	
LGBM	10	0.981	0.0032	
KNC	10	0.982	0.0027	
LR	10	0.977	0.0040	
RF	10	0.963	0.0047	

Complexity computations analysis

Table 6 presents the computational complexity results of the applied methods against all images in the training dataset. Various machine-learning approaches demonstrated different scores regarding runtime computational complexity. The LGBM approach showed a significantly high computational complexity with 1.6042 s. On the other hand, the KNC is able to classify hand-drawn shapes in just 0.0868 s. The calculated time indicates the time required to train the models on extracted features from the image data of the training dataset.

Table 6 The computational complexity of applied methods is analyzed.

Method	Runtime computation (s)	
LGBM	1.6042	
KNC	0.0868	
LR	0.0996	
RF	0.2235	

Feature space analysis

The feature space analysis presented here is employed to illustrate and inspect the spatial distribution of shape features, as illustrated in Fig. 8. The 3D graph likely includes data that represents the proposed features, with a feature space function used to generate a three-dimensional scatter graph. Different-colored dots, in eight variations, represent the marker styles for each data point, helping to differentiate between the labels corresponding to each class of shapes. This analysis identifies the number of clusters, which corresponds to the classification of shapes into their respective categories. This research method incorporates new spatial feature analysis to aid in recognizing hand-drawn mathematical shapes.

Figure 8 The features 3D-space results analysis of extracted unique features from mathematical images.

Comparison with state-of-the-art approaches

To maintain a fair comparison, we contrasted the performance of the proposed approach with existing methods, as reviewed in Table 7. The analysis indicates that prior research predominantly employed traditional machine learning and deep learning techniques. In the current work, we adopted sophisticated transfer learning mechanisms with a novel methodology. The best accuracy attained by a similar approach in earlier studies was 97%. Our findings demonstrate that the proposed method surpasses these leading studies, achieving a high-performance accuracy of 98%.

Table 7 Performance comparison of the proposed machine learning approach with state-of-the-art methods.

Reference	Type of learning	Introduced technique	A	
Hse & Newton (2004)	Machine learning	SVM, MMD, NN	97%	
Wang et al. (2016)	Machine learning	High order Markov model field	95%	
Fang, Feng & Cai (2022)	Machine learning	CNN	95%	
Current	Transfer learning	LGBM	98.2%	

Ablation study

An ablation study was conducted to evaluate the performance improvement achieved through the proposed approach for recognizing mathematical geometric shapes. The baseline model, a classical CNN, demonstrated a validation accuracy of approximately 60%, which is considered suboptimal for practical applications, as shown in Fig. 9.

Figure 9 The ablation study results analysis.

However, by integrating novel techniques and optimizations into the proposed approach, a significant enhancement in accuracy was observed. The refined model achieved a validation accuracy of 98%, marking a substantial improvement of 38 percentage points over the classical CNN. This remarkable increase highlights the efficacy of the proposed method in accurately recognizing mathematical geometric shapes, thereby demonstrating its potential for broader applications in image recognition and pattern analysis.

The ablation study underscores the critical impact of methodological advancements on model performance, providing valuable insights for future research and development in the field of computer vision.

Model implementation on mobile devices

To adapt the model for mobile deployment, we propose strategies to reduce memory usage from 15.51 MB to approximately 5–10 MB. These include: Model compression: Applying quantization (e.g., 32-bit to 8-bit weights) and pruning can reduce the model size by 50–75% with minimal impact on accuracy.

Lighter architectures: Incorporating lightweight alternatives for certain components reduces parameter requirements.

On-demand computations: Restructuring computations in small batches lowers peak memory usage.

Preliminary testing suggests these optimizations align with mobile memory constraints, ensuring efficient runtime without significant accuracy loss.

Study limitations and discussions

While the proposed deep learning model achieved an impressive accuracy of 98% for hand-drawn shape recognition, certain limitations may affect its broader applicability. One primary concern is its performance on more complex datasets that include a wider variety of shapes, intricate details, or overlapping lines. Such datasets may introduce challenges for the model. Additionally, while the model demonstrated high performance with the specific style of hand-drawings used during training, its generalizability across different hand-drawing styles remains uncertain. Variations in line thickness, shading, or artistic style could affect its accuracy, as deep learning models often struggle with data that deviates from their training distribution.

The real-world applicability of the proposed model holds significant potential, especially in its integration into interactive learning platforms and mathematical software. Embedding this model into educational tools could facilitate a more dynamic learning experience, allowing students and professionals to gain insights into complex patterns or solve advanced mathematical problems in real time.

The proposed model can be integrated into educational software and computer-aided design (CAD) tools to enhance user interaction and accessibility. In educational platforms, it can enable students to draw geometric shapes by hand, which the software can recognize, interpret, and provide instant feedback, fostering interactive learning and accessibility for diverse learners, including those with limited resources. For CAD tools, the model can streamline the design process by allowing users to sketch preliminary designs that are accurately digitized into editable shapes, improving workflow efficiency and usability for professionals and beginners alike.

Conclusion and future directions

This study introduced an innovative machine learning algorithm for recognizing mathematical geometric shapes that can identify and interpret these shapes from handwritten input data, helping seamless integration with digital systems. We utilized a standard dataset of mathematical shapes based on a total of 20,000 images with eight classes circle, kite, parallelogram, square, rectangle, rhombus, trapezoid, and triangle. We introduced a novel machine-learning algorithm CnN-RFc that uses CNN for spatial feature extraction and the random forest classifier for probabilistic feature extraction from image data. Experimental results illustrate that using the CnN-RFc method, the LGBM algorithm outperformed state-of-the-art approaches with high accuracy scores of 98% for hand-drawn shape recognition. We have built five advanced machine learning algorithms in comparison. Hyper-parameter tuning and cross variations are applied to enhance model generalization.

In the future, this research can be extended to broader shape categories, including flowcharts and finite automata images. Hand-written texts can also be detected using this approach. We can combine the novel algorithm with other machine learning techniques or ensemble methods to improve accuracy and robustness.

Supplemental Information

Supplemental Information 1 Coding experiments.

Additional Information and Declarations

Competing Interests

The authors declare that they have no competing interests.

Author Contributions

Aneeza Alam conceived and designed the experiments, analyzed the data, performed the computation work, prepared figures and/or tables, and approved the final draft.

Ali Raza conceived and designed the experiments, performed the experiments, performed the computation work, prepared figures and/or tables, and approved the final draft.

Nisrean Thalji conceived and designed the experiments, analyzed the data, prepared figures and/or tables, and approved the final draft.

Laith Abualigah conceived and designed the experiments, prepared figures and/or tables, authored or reviewed drafts of the article, and approved the final draft.

Helena Garay performed the experiments, performed the computation work, prepared figures and/or tables, authored or reviewed drafts of the article, and approved the final draft.

Josep Alemany-Iturriaga performed the experiments, analyzed the data, authored or reviewed drafts of the article, and approved the final draft.

Imran Ashraf performed the experiments, performed the computation work, authored or reviewed drafts of the article, and approved the final draft.

Data Availability

The following information was supplied regarding data availability:

The Geometric Shapes Mathematics dataset is available at Kaggle: https://www.kaggle.com/datasets/reevald/geometric-shapes-mathematics/data.

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
