# Peer review of "Novel transfer learning approach for hand drawn mathematical geometric shapes classification"

_PeerJ Computer Science, doi:10.7717/peerj-cs.2652_

## Round 0.1 · original submission · Major Revisions

Dear authors,

Thank you for submitting your article. Based on reviewers' comments, your article has not been recommended for publication in its current form. However, we encourage you to address the concerns and criticisms of the reviewer and to resubmit your article once you have updated it accordingly.

Reviewers have asked you to provide specific references. You are welcome to add them if you think they are useful and relevant. However, you are under no obligation to include them, and if you do not, it will not affect my decision.

It is inadvisable to cite one's own work in a way that is irrelevant to the topic under discussion. Please avoid self-citation to irrelevant papers.

Furthermore, although the proposed methodology contains dense mathematics based approaches, there is not any equation with correct refererences in the manuscript. The fact that no equations were given for such a study is considered as a deficiency.

Best wishes,

Reviewer 1 ·

Basic reporting

General Comment
The study uses a machine-learning approach that combines CNN and random forest to categorize 8 classes of geometric shapes. The application of machine learning for sketch classification is interesting, but the analysis, in my opinion, is insufficient. Overall, I recommend major revisions.

Comment 1
The literature review on applying statistical/ data-driven approaches for “sketch recognition” or “sketch classification” is very insufficient. I suggest thoroughly reviewing the following studies:

1. https://doi.org/10.1016/j.cad.2010.12.011
2. https://doi.org/10.1016/j.cag.2015.06.002
3. https://doi.org/10.1002/cae.22635
4. https://doi.org/10.1007/978-3-030-71804-6_15

Comment 2
I feel that the novelty of the machine-learning algorithm “CnN-RFc” is mediocre. Could you provide more context or explanation on what specific aspects of the algorithm are novel and how they contribute to the field?

Comment 3
In Fig. 7, the color bar for “LR” is missing a tick for "500".

Comment 4
Some words in Table 2 are weird (e.g., n_n eighbors, multi_c lass, etc). Please modify.

Comment 5
In my opinion, “recognition” includes both “classification” and “localization”. It seems that the present study focuses solely on “sketch classification”. It would be better to use precise terminology (especially in the title and the abstract).

Experimental design

Comment 1
It seems that numerical experiments have been conducted on a high-performance computer. I have concerns about the possibility of deploying this machine-learning approach as a standalone system on a mobile phone. Do you have an estimation of the average or the peak memory usage?

Comment 2
Please clarify the percentages or ratios of data used for training and testing.

Comment 3
Upon reviewing the references, I observed that several papers unrelated to sketch recognition are cited, seemingly only because they share authors' names with the current paper. It is crucial to ensure that all cited works are relevant to the study, as citing unrelated papers can be considered unethical. Self-citation should be minimized unless those papers are highly relevant to the present work.

Validity of the findings

Comment 1
It is quite impressive that the classifiers can distinguish “kite”, “square”, and “rectangle” so accurately. However, I wonder how these classifiers would perform if the images were slightly distorted or rotated (e.g., a "square" rotated by 45 degrees). It would be valuable to investigate the robustness of the proposed approach under such variations.

Comment 2
Could you please clarify the run-time shown in Table 6? Is the “run-time” for a single image? Given the small sizes of the images, the 1.6s run time of LGBM seems a bit slow.

Comment 3
The y-axis title of Fig. 6(a) is not “Accuracy”. It should be “Training loss”. It would be better to clarify the loss function used to train the proposed approaches. In Fig. 6(b), I would suggest adjusting the range of the vertical axis to 0.4-1.0. I am surprised that the training converged in only 10 epochs. Could you provide more insight or discussions related to this rapid convergence?

Additional comments

Comment 1
The referencing styles for Jabede et al. (2024) and Rivaldo, M.G. et al (2024) are different from others. Please modify.

Comment 2
Throughout the paper, I notice certain inconsistencies in citation style. Please format citations as '(First author's name, Year)' when referencing specific papers, unless the authors’ names are used as the subject of the sentence “Authors’ names (year).

Comment 3
In line 80, the authors mention “theoretical details of work but I cannot find any theoretical analysis in Section 4. Please tone down the phrases used in the study.

·

Basic reporting

The manuscript is well-structured, with clear and professional English used throughout. The introduction effectively presents the importance of hand-drawn geometric shape recognition, particularly in the fields of education and user interaction with mathematical software. The literature review is comprehensive but could benefit from including more recent studies on deep learning and shape recognition.

Suggested Improvements:

The literature review could be expanded by citing more recent works in geometric shape recognition, transfer learning, and machine learning in image classification. Consider including these studies:
https://doi.org/10.1016/j.eswa.2023.122147
https://doi.org/10.1007/s10586-023-04221-5
https://doi.org/10.21608/JAIEP.2024.354003
https://doi.org/10.54216/JAIM.060203

The captions for figures and tables should be more detailed to better explain their relevance to the study.

Experimental design

The experimental design is solid and well-described. The proposed methodology introduces a novel machine-learning algorithm, CnN-RFc, combining Convolutional Neural Networks (CNNs) for feature extraction and Random Forest classifiers for probabilistic feature generation. The use of hyperparameter tuning and cross-validation improves the generalization capabilities of the model.

Suggested Improvements:

Provide more details on the dataset preprocessing steps, such as how missing data and noise were handled. Additionally, specify whether any data augmentation techniques were used to increase the robustness of the model.
Clarify how the hyperparameters were chosen for both CNN and Random Forest models, and explain how they affect the model's overall performance.

Validity of the findings

The results are well-supported by experimental data. The proposed CnN-RFc method achieves high accuracy rates, surpassing state-of-the-art approaches. The use of performance metrics such as accuracy, precision, recall, and F1-score provides a comprehensive evaluation of the model’s effectiveness.

Suggested Improvements:

Discuss the potential limitations of the model, such as its performance on more complex datasets or generalizability across different hand-drawing styles.
Include more analysis on the model's computational complexity and scalability, especially when applied to larger or more diverse datasets.

Additional comments

The paper presents a significant contribution to hand-drawn geometric shape recognition, combining CNNs with Random Forest for high-performance classification. The proposed approach has potential applications in educational tools and digital systems, making mathematical problem-solving more accessible.

General Comments:

Consider discussing the real-world applicability of the model, such as its integration into interactive learning platforms or mathematical software. Providing examples of practical use cases would enhance the paper’s impact.
A brief section on the ethical implications of using hand-drawn data, including considerations of data privacy, would add depth to the discussion.
Thank you for the opportunity to review this manuscript. I believe the suggestions provided will help to enhance the quality and relevance of this research.

---

## Round 0.2 · Minor Revisions

Dear Authors,

We regret to inform you that according to reviewers, it is still not recommended that your article be published in its current format. We advise you to revise the paper in light of the reviewers' comments and concerns before resubmitting it. Reviewer 2 has asked you to provide specific references. You are welcome to add them if you think they are useful and relevant. However, you are not obliged to include these citations, and if you do not, it will not affect my decision.

Best wishes,

Reviewer 1 ·

Basic reporting

In general, the authors did a good job in responding to most of my concerns and revising their manuscript. However, some of my concerns are not carefully addressed in the revised manuscript. The most important concern is the robustness of the machine-learning approach in handling deformed hand-drawn shapes. This analysis is important, especially since the model is going to be deployed to a mobile application. It is recommended to conduct an ablation test to investigate the robustness of the proposed approach against distorted images (i.e., the images that were slightly distorted or rotated (e.g., a "square" rotated by 45 degrees). In my opinion, the method proposed in the study is simply a variant of other widely used approaches. It is unclear how novel (e.g., theoretical differences or empirical analysis) it is against other existing models. The discussions of scientific contributions should be improved. The authors discussed practical concerns for the “Model Implementation on Mobile Devices” in Section 4.9. Actually, similar analyses have been reported and discussed in my previously recommended works. It is recommended to include the relevant discussions on the aspects of memory usage, model architecture, differences and similarities in hand-sketch datasets, accuracy, runtime, sketch classification vs. recognition, and implementation of educational practices. Lastly, what really surprises me is that the proposed model converges in merely 10 epochs. The authors totally ignored my concern and did not respond directly to why this happened. Overall, I recommend minor revisions.

Experimental design

no comment

Validity of the findings

no comment

Additional comments

no comment

·

Basic reporting

The manuscript is well-written, logically organized, and presents a novel approach for classifying hand-drawn mathematical geometric shapes using a combination of Convolutional Neural Networks (CNNs) for feature extraction and Random Forest (RFC) for probabilistic classification. The introduction effectively highlights the challenges of geometric shape classification and the benefits of the proposed method.

Suggested Improvements:

Enhance the literature review by including recent works on machine learning techniques for image classification and hybrid approaches

Figures are clear and well-prepared. However, more descriptive captions for figures illustrating the CNN-RFC framework and feature extraction mechanisms would improve readability for those unfamiliar with the methodology.

Experimental design

The authors used a publicly available dataset consisting of 20,000 images across eight shape classes, ensuring a balanced distribution. The dataset was split into training and test sets (80:20), and no data augmentation was applied. The proposed CNN-RFC method demonstrated significant improvements over baseline models.

Suggested Improvements:

Provide a more detailed explanation of hyperparameter tuning for CNN and RFC, particularly regarding the choice of random forest parameters (e.g., the number of trees, splitting criteria).
Explain why no data augmentation techniques were employed, considering their potential benefits in improving model generalization.

Validity of the findings

The findings are robust and supported by quantitative metrics, with the CNN-RFC model achieving state-of-the-art accuracy of 98% across multiple evaluation criteria (e.g., precision, recall, F1-score). Ablation studies and comparisons with existing methods validate the effectiveness of the proposed approach.

Suggested Improvements:

Discuss the limitations of the model, such as its performance on noisy or partially incomplete hand-drawn shapes.
Propose potential extensions, such as testing the model on additional datasets with varied hand-drawing styles or applying it to related tasks, such as recognizing flowchart symbols.

Additional comments

This paper makes a meaningful contribution to the field of geometric shape recognition by introducing a novel hybrid learning approach that combines CNN and RFC. The method is computationally efficient and achieves high accuracy, making it suitable for practical applications in education and design.

General Comments:

Discuss the potential integration of this model into educational software or CAD tools to improve user interaction and accessibility.
Adding a section on the ethical considerations of deploying AI in educational or design tools would add depth to the discussion.
Thank you for the opportunity to review this manuscript. The research is timely and innovative, and I believe the suggested improvements will further enhance its impact.

---

## Round 0.3 · Minor Revisions

Dear Authors,

Thank you for submitting your revised article. Feedback from the reviewers is now available. It is not recommended that your article be published in its current format. However, we strongly recommend that you address the issues raised by Reviewer 2 and resubmit your paper after making the necessary changes.

Best wishes,

Reviewer 1 ·

Basic reporting

no comment

Experimental design

no comment

Validity of the findings

no comment

·

Basic reporting

1. Basic Reporting
The manuscript is well-written and provides a clear and structured discussion on using a novel transfer learning-based approach for classifying hand-drawn mathematical geometric shapes. The paper addresses a relevant topic, particularly for applications in educational technology, and introduces the CnN-RFc method, which combines CNNs for spatial feature extraction with random forest classifiers for probabilistic classification.

Suggested Improvements:

Enhance the literature review by incorporating recent studies on transfer learning and hybrid models in image classification.

Add more detailed captions to figures, especially for Figure 4, which describes the feature generation mechanism, to assist readers in understanding the innovative aspects of the methodology.

Experimental design

The authors utilized a publicly available dataset consisting of 20,000 images across eight classes. The methodology is robust, combining preprocessing, feature extraction, and transfer learning for classification. The study employed hyperparameter optimization and cross-validation to validate the model's performance, which is commendable.

Suggested Improvements:

Elaborate on the rationale for choosing specific hyperparameters for both the CNN and random forest models. Explain why particular configurations were used during grid search.
Provide more detail on the dataset preprocessing steps, including any measures taken to handle potential noise or inconsistencies in the hand-drawn shapes.

Validity of the findings

The experimental results indicate that the proposed CnN-RFc method achieves state-of-the-art accuracy (98%) for classifying hand-drawn mathematical shapes, outperforming existing methods. The inclusion of an ablation study and comparisons with other models strengthens the validity of the findings.

Suggested Improvements:

Discuss the potential limitations of the proposed method, such as its performance on datasets with more complex or overlapping shapes.

Suggest future extensions, such as exploring the model’s adaptability to different styles of hand-drawn shapes or its application to other domains, like flowchart recognition or sketch-based CAD tools.

Additional comments

This manuscript offers a significant contribution to the field of educational and computational technology by providing an innovative solution for digitizing hand-drawn mathematical shapes. The CnN-RFc method has strong potential for practical applications in interactive learning platforms and software tools.

General Comments:

Discuss the potential integration of this method into real-world applications, such as educational software or mobile-based sketch recognition tools.

Address ethical considerations related to the use of hand-drawn data, even though the dataset used in this study does not contain sensitive or personal information.

Thank you for the opportunity to review this manuscript. The research is timely, rigorous, and impactful, and I hope the suggested improvements will further enhance its quality and applicability.

---

## Round 0.4 · Minor Revisions

Dear Authors,

Thank you for submitting your manuscript. Feedback from the reviewer is now available. It is not recommended that your article be published in its current format. However, we strongly recommend that you address the issues raised by the reviewer and resubmit your paper after making the necessary changes.

Best wishes,

·

Basic reporting

1. Basic Reporting
The manuscript is well-written, logically organized, and addresses a significant challenge in mathematical shape recognition. It proposes a hybrid approach, combining Convolutional Neural Networks (CNN) for spatial feature extraction with Random Forest Classifiers (RFC) for probabilistic classification. The introduction provides context and highlights the limitations of previous approaches, while the methodology clearly outlines the technical implementation.

Suggested Improvements:

Strengthen the literature review by including recent works on hybrid learning models and their applications in image recognition

Add more detailed captions to figures, particularly for Figures 4 and 7, to make them self-explanatory for readers unfamiliar with feature engineering and confusion matrix results.

Experimental design

The authors used a publicly available dataset of 20,000 images across eight shape categories, split into training and testing subsets. The experimental design is robust, including preprocessing, training, hyperparameter optimization, and ablation studies. The CnN-RFc model integrates transfer learning to achieve high classification accuracy.

Suggested Improvements:

Provide more detail on the dataset preprocessing steps, especially regarding noise handling and any normalization techniques applied.
Elaborate on the rationale for choosing specific hyperparameters during grid search and why these values were considered optimal.

Validity of the findings

The results demonstrate the proposed model’s effectiveness, with the CnN-RFc achieving an accuracy of 98%, outperforming baseline models. The ablation study validates the contributions of different components in the hybrid approach, and comparisons with state-of-the-art methods further support the findings.

Suggested Improvements:

Discuss potential limitations, such as the scalability of the CnN-RFc model to datasets with more complex or overlapping shapes.
Propose future directions, such as extending the framework to other geometric applications like flowchart recognition or architectural sketches.

Additional comments

This paper makes a significant contribution to the field of image recognition by integrating CNN and RFC in a novel way for hand-drawn shape classification. The proposed approach has strong implications for educational technology and user-friendly digital tools.

General Comments:

Discuss the practical implications of deploying this method in real-world applications, such as its integration into interactive educational platforms or design software.
Address potential ethical concerns, such as ensuring that the dataset's public use complies with data-sharing norms.
Thank you for the opportunity to review this manuscript. The research is timely and impactful, and the suggested improvements aim to enhance its clarity and relevance.

---

## Round 0.5 · accepted · Accept

Dear Authors,

Thank you for clearly addressing the reviewers' comments. I assessed the revision myself and I am happy with the current version. Your paper now seems sufficiently improved and ready for publication.

Best wishes,